# A Retrospective Medical Record Review of Adults with Non-Cancer Diagnoses Prescribed Medicinal Cannabis

**DOI:** 10.3390/jcm12041483

**Published:** 2023-02-13

**Authors:** Michael Morris, Richard Chye, Zhixin Liu, Meera Agar, Valentina Razmovski-Naumovski

**Affiliations:** 1South West Sydney Clinical Campuses, Faculty of Medicine & Health, University of New South Wales Sydney (UNSW), Sydney, NSW 2170, Australia; 2Sacred Heart Health Service, St Vincent’s Hospital, Darlinghurst, NSW 2010, Australia; 3Stats Central, University of New South Wales Sydney (UNSW), Sydney, NSW 2170, Australia; 4Ingham Institute for Applied Medical Research, Liverpool, NSW 2170, Australia; 5Improving Palliative, Aged and Chronic Care through Clinical Research and Translation (IMPACCT), Faculty of Health, University of Technology Sydney, Ultimo, NSW 2007, Australia

**Keywords:** medicinal cannabis, conditions, indications, non-cancer, pain, retrospective review, symptoms

## Abstract

Research describing patients using medicinal cannabis and its effectiveness is lacking. We aimed to describe adults with non-cancer diagnoses who are prescribed medicinal cannabis via a retrospective medical record review and assess its effectiveness and safety. From 157 Australian records, most were female (63.7%; mean age 63.0 years). Most patients had neurological (58.0%) or musculoskeletal (24.8%) conditions. Medicinal cannabis was perceived beneficial by 53.5% of patients. Mixed-effects modelling and post hoc multiple comparisons analysis showed significant changes overtime for pain, bowel problems, fatigue, difficulty sleeping, mood, quality of life (all *p* < 0.0001), breathing problems (*p* = 0.0035), and appetite (*p* = 0.0465) Symptom Assessment Scale scores. For the conditions, neuropathic pain/peripheral neuropathy had the highest rate of perceived benefit (66.6%), followed by Parkinson’s disease (60.9%), multiple sclerosis (60.0%), migraine (43.8%), chronic pain syndrome (42.1%), and spondylosis (40.0%). For the indications, medicinal cannabis had the greatest perceived effect on sleep (80.0%), followed by pain (51.5%), and muscle spasm (50%). Oral oil preparations of balanced delta-9-tetrahydrocannabinol/cannabidiol (average post-titration dose of 16.9 mg and 34.8 mg per day, respectively) were mainly prescribed. Somnolence was the most frequently reported side effect (21%). This study supports medicinal cannabis’ potential to safely treat non-cancer chronic conditions and indications.

## 1. Introduction

Cannabis (Cannabaceae) has been used medicinally since 400 AD for its analgesic, appetite enhancement, and myorelaxant properties [1,2,3]. Emerging evidence suggests that people with chronic conditions may benefit from using medicinal cannabis for treating chronic pain, multiple sclerosis-related spasticity, epilepsy, Parkinson’s disease, insomnia, and anxiety [4]. However, evidence supporting medicinal cannabis for these conditions/symptoms is generally anecdotal and inconclusive [5]. Limitations include short study durations, small sample sizes, differing formulations/modes of administration, lack of follow-up [4], and stigma surrounding its addiction and abuse [6].

In Australia, impressive public support instigated legislative changes in 2016 allowing medical doctors to prescribe cannabis-based medicines for chronically ill patients subject to the approval from the Therapeutic Goods Administration [7] which is the medicine and therapeutic regulatory agency of the Australian Government. Unregistered cannabis-based medicines can be prescribed through the TGA Special Access Scheme, in particular, Category B (SAS Cat-B) [8], and this requires clinical justification for patient use, reasons why registered medications have not been appropriate, and the expected benefits versus harms of the proposed treatment; or Authorized Prescriber Program (using medicines with an established history of use) [9]. In addition, doctors will assess any problems with drug dependence and substance abuse. In Australia, there has been exponential growth in the use of medicinal cannabis: by 2021, over 333,000 SAS Cat-B approvals had been issued to patients and over 1807 authorized prescribers, which shows the growing interest in legally prescribed medicinal cannabis [8,9].

Rapidly increasing public interest in medicinal cannabis necessitates research into the conditions/symptoms it can potentially treat, and its safety profile and this has prompted the Australian government to fund clinical trials to ascertain its place in mainstream medicine [10]. Whilst we wait for these results, retrospective studies assist in understanding benefits and harms in real-world clinical practice and provide data to guide future clinical trials [11]. There are few retrospective studies that report on all facets of medicinal cannabis use including demographics, perceived efficacy, cannabis formulations, and side effects.

This retrospective study will aim to answer the questions: (1) What are the characteristics of people with non-cancer diagnoses using medicinal cannabis? and (2) Does medicinal cannabis provide effective relief for patient conditions and symptoms? Secondary objectives include determining dosage, side effects, and reasons for cessation of treatment.

## 2. Methods

### 2.1. Study Design, Setting, and Population

A retrospective medical record review was conducted at Wolper Jewish Hospital in Sydney, Australia, which is a stand-alone, not-for-profit clinic specializing in rehabilitation and palliative care. A consent form was provided which talks about the risk of psychosis and schizophrenia. Medical records of patients prescribed medicinal cannabis between 1 February 2018 and 30 November 2021 were reviewed using the inclusion criteria: (1) 18 years or older; and (2) non-cancer diagnosis. This study was approved by the St Vincent’s Hospital Sydney Human Research Ethics Committee (Approval No: 2020ETH00008).

### 2.2. Characteristics Retrieved

Patient demographic data (age, gender, postcode, country of birth, indigenous status, and marital status), medical data (condition(s), primary and secondary indication(s)), administration route, cannabinoid ratio, dose, number of cannabis-based medicine(s), and previous cannabis use were recorded. Side effects, treatment continuation, and reasons for cessation were quantitatively described.

### 2.3. Evaluation of Effectiveness

The effectiveness of medicinal cannabis treatment was examined by the following: (1) The patient’s own perceived benefit reported as ‘beneficial’, ‘not beneficial’ or ‘unclear’; this data was used as a proxy to determine overall and condition-specific perceived benefit. (2) The Palliative Care Outcomes Collaboration (PCOC) Symptom Assessment Scale (SAS) scores. This validated tool reliably measures patient-reported symptom distress amongst those with advanced disease and includes pain, fatigue, breathing problems, bowel problems, appetite, nausea, difficulty sleeping, and mood. Each symptom is rated out of 10 (0 = not at all to 10 = worst possible). Scores 0–3 depict mild distress, 4–7 moderate distress, and 8–10 severe distress [12]. Patients also rated their quality of life (QoL) out of 10 (0 = worst possible to 10 = best possible). Pre-treatment SAS scores were recorded at 5–6 weeks, 3, 6, 9, 12 months, etc., depending on patients’ clinic attendance.

### 2.4. Statistical Analysis

De-identified patient data were entered into an Excel spreadsheet and analyzed descriptively. To account for missing data, patient scores for each SAS symptom were grouped into baseline and three-month blocks, averaged, and analyzed by fitting a mixed-effects model [13]. Post hoc multiple comparisons analysis using Dunnett’s test identified the statistical significance of timepoints compared to baseline. Analysis was stopped at the timepoint which had around 50% of the initial patient cohort for each condition or symptom. Cohorts with low patient numbers were analyzed with a Wilcoxon signed-rank test. Two-tailed *t*-test compared the highest doses of cannabidiol (CBD) and delta-9-tetrahydrocannabinol (Δ-9-THC). Statistical significance was set at *p* < 0.05. Statistical tests were performed using GraphPad Prism (v9.1.2, GraphPad Software Inc., San Diego, CA, USA).

## 3. Results

### 3.1. Characteristics of the Included Patients

Two-hundred-and-seven patients met the study inclusion criteria. Fifty patients were excluded due to missing files/poor data, resulting in 157 patients analyzed.

Patients were predominantly female (63.7%), aged 63.0 years, and from the Greater Sydney region (86.6%). Most patients were born in Australia (66.9%) and partnered (57.3%), with 17.8% stating previous cannabis use (Table 1).

Patients reported an array of conditions (Appendix A) and indications (Appendix A) treated with medicinal cannabis. Primarily, patients (87.3%) reported one condition which fell under the neurological (58.0%) and musculoskeletal class (24.8%). The most common patient indications included pain (86.6%), muscle spasm (11.5%), and sleep (6.4%) (Table 1).

Patients were mostly prescribed one cannabis-based medicine (CBM; 71.3%), an oral oil CBM (99.4%) with balanced Δ-9-THC and CBD (84.7%). The mean highest CBD dose (34.8 ± 48.04 mg/day) was significantly higher (*p* < 0.0001) than Δ-9-THC (16.9 ± 17.23 mg/day) (Table 1).

### 3.2. Effectiveness of Treatment—Patient-Perceived Benefit

Just over half of patients (53.5%) perceived medicinal cannabis as being ‘beneficial’ regardless of their condition/indication (Figure 1a). Patients aged <65 years perceived relatively greater benefit from medicinal cannabis treatment (56%; *n* = 75) compared to those aged ≥65 (51.2%; *n* = 82) (Figure 1b).

The conditions and indications with ≥10 patients were analyzed further. For the conditions, neuropathic pain/peripheral neuropathy had the highest rate of perceived benefit (66.6%), followed by Parkinson’s disease (60.9%), multiple sclerosis (60.0%), migraine (43.8%), chronic pain syndrome (CPS) (42.1%), and spondylosis (40.0%) (Figure 1c). For the indications, medicinal cannabis had the greatest perceived effect on sleep (80.0%), pain (51.5%), and muscle spasm (50.0%) (Figure 1d).

### 3.3. Effectiveness of Treatment—Overall Symptom Assessment Scale Scores

Figure 2 shows the trends of the scores over time in 3-month blocks, with improved scores to 3 months and general stabilization of scores to 30 months. Mixed-effects modelling showed treatment with medicinal cannabis was associated with significant changes over time for all symptom distress (except ‘nausea’) and ‘QoL’ (Table 2). Post hoc analysis showed significant score improvements across timepoints compared to baseline for: ‘breathing problems’ to 6 months; ‘bowel problems’ to 9 months; and ‘pain’, ‘difficulty sleeping’, ‘fatigue’, ‘mood’, and ‘QoL’ scores to 12 months (Table 2). From the scores, ‘breathing problems’, ‘nausea’, and ‘appetite’ symptoms were not causing significant distress to patients and were not analyzed further.

### 3.4. Conditions

#### 3.4.1. Spondylosis

In patients with spondylosis, improved trends were observed for ‘fatigue’, ‘difficulty sleeping’, ‘mood’, and ‘QoL’ scores to 6 months; and ‘pain’ and ‘bowel problems’ to 9 months (Figure 3a). Mixed-effects modelling showed significant changes over time in all symptom scores but no significant improvements in QoL (Table 3). Post hoc analysis found significant decreases compared to baseline for: ‘mood’ and ‘fatigue’ to 6 months; and ‘pain’ and ‘difficulty sleeping’ to 9 months (Table 3).

#### 3.4.2. Parkinson’s Disease

In patients with Parkinson’s disease, improved trends were observed for scores to 3 months (‘QoL’ to 6 months), with general stabilization over time (Figure 3b). Mixed-effects modelling showed significant changes over time for ‘pain’, ‘fatigue’, and ‘difficulty sleeping’ scores (Table 3). Post hoc analysis revealed significant decreases compared to baseline for: ‘pain’ and ‘mood’ at 3 months; and ‘fatigue’ and ‘difficulty sleeping’ to 12 months (Table 3).

#### 3.4.3. Chronic Pain Syndrome

In patients with chronic pain syndrome, improved trends were observed for the scores to 3 months which generally continued to 9 months (Figure 3c). Mixed-effects modelling showed significant changes over time in scores for ‘pain’, ‘fatigue’, ‘difficulty sleeping’, and ‘mood’ (Table 3). Post hoc analysis found significant decreases compared to baseline for: ‘mood’ and ‘difficulty sleeping’ at 3 months, ‘fatigue’ at 9 months, ‘pain’ to 12 months, with improved ‘QoL’ scores at 3 months (Table 3).

#### 3.4.4. Neuropathic Pain and Peripheral Neuropathy

In patients with neuropathic pain/peripheral neuropathy, improved trends were observed for ‘fatigue’ and ‘difficulty sleeping’ scores to 3 months; ‘pain’ and ‘QoL’ to 6 months; ‘bowel problems’ to 9 months, and mood to 12 months (Figure 3d). Mixed-effects modelling analysis showed significant changes over time in ‘pain’, ‘fatigue’, and ‘difficulty sleeping’ scores (Table 3). Post hoc analysis showed significant decreases compared to baseline for: ‘bowel problems’ at 3 months, ‘difficulty sleeping’ to 6 months, and ‘pain’ to 12 months, with improved ‘QoL’ scores at 3 months (Table 3).

#### 3.4.5. Migraine

In patients with migraine, improved trends were observed for ‘mood’ to 3 months; ‘fatigue’, ‘bowel problems’, ‘difficulty sleeping’, and ‘QoL’ scores to 6 months, and ‘pain’ to 18 months (Figure 3e). Mixed-effects modelling showed significant changes over time for ‘fatigue’ and ‘difficulty sleeping’ scores (Table 3). Post hoc analysis showed significant decreases compared to baseline for ‘fatigue’ scores to 6 months (Table 3).

#### 3.4.6. Multiple Sclerosis

There are downward trends for ‘pain’ (initially to 3 months) and ‘bowel problems’ (initially to 6 months) scores to 12 months, and ‘fatigue’, ‘difficulty sleeping’, and ‘mood’ (all initially to 3 months) to 15 months, with improvement in ‘QoL’ to 6 months in patients with multiple sclerosis (Figure 3f). A Wilcoxon signed-rank test showed that the ‘fatigue’, ‘difficulty sleeping’, and ‘mood’ scores were significantly decreased at 3 months compared to baseline (Table 3).

### 3.5. Indications

#### 3.5.1. Pain

In patients with pain, improved trends were observed for all symptom scores to 3 months (‘QoL’ to 6 months) with general stabilization over time (Figure 4a). Mixed-effects modelling showed significant changes over time for all symptoms scores (Table 4). Post hoc analysis found significant decreases compared to baseline for: ‘bowel problems’ and ‘mood’ to 9 months; and ‘pain’, ‘fatigue’, and ‘difficulty sleeping’ to 12 months, with improved ‘QoL’ scores to 12 months (Table 4).

#### 3.5.2. Muscle Spasm

In patients with muscle spasm, initial improved trends were observed for all symptom scores to 3 months; ‘pain’, ‘fatigue’, and ‘bowel problems’ to 12 months; and ‘difficulty sleeping’, ‘QoL’, and ‘mood’ to 15 months (Figure 4b). Mixed-effects modelling showed significant changes over time for ‘pain’ and ‘fatigue’ scores (Table 4). Post hoc analysis showed significant decreases compared to baseline for: ‘pain’ at 3 months, ‘fatigue’ at 3, 9, and 12 months, and ‘difficulty sleeping’ to 6 months (Table 4).

#### 3.5.3. Sleep

In patients with sleep issues, improved trends were observed for ‘fatigue’, ‘bowel problems’, and ‘mood’ scores to 3 months; ‘difficulty sleeping’ to 6 months; and ‘pain’ and ‘QoL’ scores (initially to 3 months) to 9 months (Figure 4c). Wilcoxon signed-rank test showed ‘fatigue’ and ‘difficulty sleeping’ scores were significantly lower at 3 months compared to baseline (Table 4).

### 3.6. Safety and Treatment Status

The most common side effects as reported by 68 patients (43.3%, *n* = 157) were somnolence (21.0%), dry mouth (7.6%), disorientation/confusion (3.8%), intoxication (3.8%), constipation (2.5%), and dizziness (2.5%) (Figure 5a). Unsteadiness, mood swings, vivid dreams, hallucinations, delirium, paranoia, agitation, depression, erectile dysfunction, headaches, nausea, anxiety, tachycardia, and teeth grinding were experienced by three or fewer patients (≤2.5%).

Around 64.8% of patient treatments were ongoing at the time of data collection, whilst 22.3% had ceased treatment. The treatment status was unknown for 22.9% of patients (*n* = 157). Thirty-nine reasons were reported for treatment cessation, with the main being inefficacy (51.3%), followed by side effects (25.6%), and relief from other medications/treatment (12.8%) (Figure 5b). One patient died from disease complications unrelated to medicinal cannabis treatment.

## 4. Discussion

This retrospective medical record review describes the population characteristics of patients using medicinal cannabis at a clinic in Sydney, Australia and provides data on the effectiveness and safety of medicinal cannabis treatment on patient conditions and indications. In our study, the mean age of patients was higher than in international studies (40–50 years) [14,15,16,17,18,19]. In Australia, patients must have trialed mainstream therapies before their medicinal cannabis application can be approved [4]. Patients were predominantly female, consistent with previous prospective studies of non-cancer patients [14,16,20]. Most patients were from the Greater Sydney Region which agrees with another study which found 82.6% of patients lived in urban areas [21]. This is indicative of medicinal cannabis prescribers and dispensers being established in larger urban areas [22]. More than half of the patients were partnered. This may make it easier for patients to undertake daily errands, including attending medical appointments whilst on medicinal cannabis as driving with cannabis containing ∆-9-THC is illegal in Australia [23].

It difficult to ascertain the actual numbers of patients that had used cannabis before attending the clinic as patients are under no obligation to state previous cannabis use. However, the results here are similar to another Australian study which showed that 14.1% had not used cannabis before using it for medical reasons [24].

In this study, most patients were prescribed oral oil cannabis-based medicines of balanced Δ-9-THC:CBD ratios reflecting their availability for prescription in Australia [4]. This could be due to cannabis flower being very expensive and the ease of administrating oils compared to vaporization [4]. In contrast, international patients predominantly inhale cannabis; however, oral preparations are becoming more prevalent due to greater physician advocation [15,16,21,25].

Literature shows that therapeutic doses of Δ-9-THC (5–20 mg) tend to be smaller than CBD (50–1500 mg) [4] and our study supports this finding. This is expected as CBD is non-psychoactive and may circumvent the side effects of Δ-9-THC [17]. Our study showed that higher doses of Δ-9-THC were used which may indicate that people with non-cancer conditions/indications may be able to tolerate higher doses compared to people with cancer, and this needs to be ascertained in clinical use [19].

Most patients in our study sought treatment for neurological and musculoskeletal conditions, with pain as the main indication. This agrees with data indicating that most approvals in Australia are for chronic pain, and pain ranks as the number one indication in several studies [4,14,16,21]. In previous studies, the frequency of insomnia and psychiatric disorders is higher [16,18,21], with one study reporting 9.7% and 27% of patients, respectively [26]. Our study’s lower incidence (1.3% and 0.6%, respectively) can be explained by the prescribing medical doctor referring patients with sleep and mental health disorders to other specialists to investigate/rule out specific causes of these conditions and measure their severity.

The presence of symptoms such as pain, fatigue, sleep difficulties, and poor mood in this study indicates high symptom burden across these chronic diseases [27,28]. For overall effectiveness, just over half of the patients believed medicinal cannabis to be ‘beneficial’, with mixed-effects modelling of all patients showing statistically significant improvements in all SAS scores except for nausea.

When comparing age groups, 56% of patients under <65 years found medicinal cannabis to be beneficial compared to 51.2% of patients aged ≥65. A study of patients with cancer indicated that younger age was associated with the therapeutic success of medicinal cannabis, which could indicate more tolerance to higher doses, and this may be reflected in the present study [29].

In terms of patient-perceived benefit, the results suggest that medicinal cannabis is more effective at treating neuropathic pain conditions compared to nociceptive pain conditions such as spondylosis (the lowest of all conditions/indications), findings congruent with available evidence on different pain conditions [11,30,31,32]. A prospective observational study in patients with chronic pain showed significant score reductions in measures of pain severity and interference in daily life over a 12-month study duration compared to baseline (*p* < 0.001), as well as significant improvements in QoL (*p* < 0.05) [14]. These results are supported by other studies which demonstrated significantly improved pain scores and QoL over time [15,16]. Within the conditions, SAS pain scores in our study showed significant pain decreases over time for spondylosis, chronic pain syndrome, and neuropathic pain/peripheral neuropathy to at least 9 months. Moreover, for pain as an indication, there were sustained significant decreases in all SAS scores to at least 9 months compared to baseline which shows an overarching effect of medicinal cannabis on other symptoms. Due to the many different pain etiologies, it is difficult to ascertain pain location and/or patient’s response and tolerance to pain [33].

Medicinal cannabis was beneficial to around 60% of patients with Parkinson’s disease. A systematic review of the effects of cannabis on Parkinson’s disease, including RCTs and non-RCTs, reported improvements in sleep and pain, which agree with our findings [34]. 

For migraine, our study revealed that under half of the patients believed treatment to be beneficial, with significant SAS scores for ‘fatigue’ and a downward trend for ‘difficulty sleeping’. One retrospective chart review analyzing migraine showed that 85.1% of patients reported a decrease in mean migraine headache frequency from 10.4 to 4.6 at follow-up (*p* < 0.0001) [25]. Similarly, archival data from an app showed migraine ratings reduced by 50% (*p* < 0.001) [35]. Another retrospective review showed improvements in headache profile, sleep, and mood [36]. 

Symptoms such as fatigue, pain, sleep, and mood disorders are also prevalent in multiple sclerosis [37]. In this study, 60.0% of patients with multiple sclerosis believed treatment to be beneficial, with symptoms of ‘pain’, ‘fatigue’ ‘difficulty sleeping’, and ‘mood’ as possible targets for medicinal cannabis treatment. For the indication muscle spasm, these results were comparable. A recent review supported Sativex’s^®^ (an oral spray of balanced Δ-9-THC and CBD) effectiveness on multiple sclerosis pain and spasticity, with improvements in sleep and QoL [38]. Another prospective study found 70.5% of patients reported a ≥20% improvement in symptoms after one month [39], which agrees with our results. As Sativex was not widely used in our study, it is difficult to compare these findings to our study as predominantly oil formulations were used. However, over 80% of patients in this study used balanced formulations and our results could mirror these findings.

Poor sleep is an important symptom in chronic disease as it coincides with poor symptom control and impacts QoL [40]. Many studies treat sleep as a secondary outcome [41], with a literature review suggesting that cannabis could improve sleep in patients with chronic pain [42]. Although results for patients with sleep as an indication showed the greatest rate of perceived benefit of all indications/conditions analyzed, this data should be treated with caution due to a small sample size. Studies have shown insufficient and conflicting evidence for medicinal cannabis use in sleep disorders and suggest that the cannabinoid type, combination, dose, and route of administration need to be considered when designing RCTs [42,43]. Comparatively, our results showed significant improvement in SAS scores for ‘difficulty sleeping’ for spondylosis to 9 months and Parkinson’s disease to 12 months, with improvements in other conditions to at least 3 months, with a downward trend for migraine. This also occurred with ‘pain’ and ‘fatigue’ (and sometimes ‘mood’) in the listed conditions and indications which shows that tackling one symptom with medicinal cannabis may reciprocate on another.

Medicinal cannabis was generally safe as shown by the follow-up SAS scores to 12 months for many of the conditions/indication. Moreover, dosing was tailored to each patient (“start low, go slow” approach) by the prescribing medical doctor, thus minimizing dose-related toxicities [44]. A scoping review found that minor side effects were reported including somnolence, dizziness, and dry mouth which are congruent to this study’s findings [45]. In our study, the most reported reason for treatment cessation was perceived medicinal cannabis inefficacy followed by side effects, with our findings congruent to a prospective clinical trial [20].

The strength of this study is that patient data up to 30 months can be assessed which is longer than prospective and clinical trials [14,15]. Although missing data is unavoidable [46,47], mixed-effects modelling allowed for a valid analysis whilst reducing associated biases [13,47].

One limitation relates to the location of the study (undertaken at one clinic) which can introduce locality bias, and retrospective design of the study which limits the ability to evaluate the causality of medicinal cannabis use and improvements in symptom scores. Confounding factors may have influenced patient SAS scores. For example, a patient with spondylosis may have a higher score for their pain distress due to another developing issue. Whilst the PCOC SAS can accurately and reliably measure symptom distress in a palliative population, it lacks specific measures for many condition-specific indications e.g., mobility for Parkinson’s disease [48], number of headaches experienced by patients with migraine, etc. Moreover, SAS scores of bowel problems, mood, and QoL did not significantly improve with medicinal cannabis over time for the conditions/indications examined in this study. This could be due to these symptoms having multifaceted constructs which are not covered in the single SAS score. Thus, using specific measures, more frequently/consistently recorded, may have led to more significant results. Other limitations included differentiating between these differing pain etiologies, interpreting results in smaller samples, unclear dosing regimen, and lack of follow-up. It is not known whether this missing data can accurately reflect greater improvement and/or sustainability over time. Although the cannabis-based medicines were mostly balanced, other formulations and their effectiveness could not be determined. In addition, this study was unable to relate dose and safety, tolerance to previous use, and adverse drug-drug interactions.

## 5. Conclusions

This study indicates that medicinal cannabis, in a balanced formulation, may address a variety of non-cancer conditions and indications concurrently and can be safely prescribed by a medical doctor. The results can be used to guide future clinical trials which could investigate the conditions and indications identified here as sensitive to treatment. It is recommended that ‘pain’ and ‘difficulty sleeping’ (which may improve fatigue) be investigated further as treating these two symptoms may have reciprocating effects on each other, on mood, and QoL. Future studies should address how cannabinoids interact in the body and with one another in differing ratios and formulations, and include relevant patient-reported outcome measures and longer-term safety studies. It is recommended that patients are assessed individually to determine whether medicinal cannabis is an appropriate treatment option, considering the associated safety risks to potential therapeutic benefit. Understanding how medicinal cannabis can be used in mainstream medicine is crucial as it has the potential to positively impact millions of lives around the world.

## Figures and Tables

**Figure 1 jcm-12-01483-f001:**
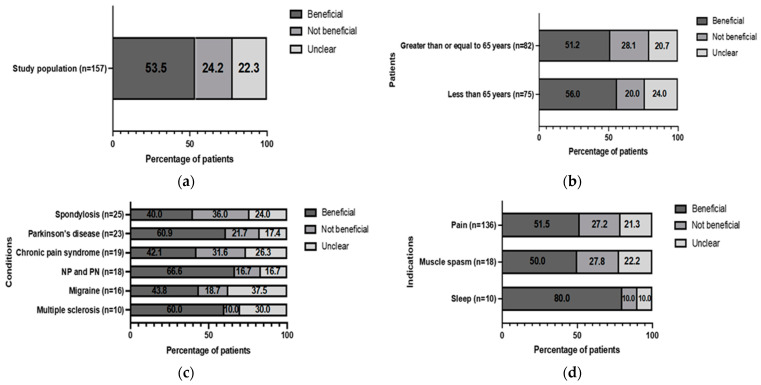
Patient-perceived benefit of medicinal cannabis treatment: (**a**) overall (regardless of condition or indication); (**b**) comparing <65 years and ≥65 years; (**c**) by condition; (**d**) by indication. NP: neuropathic pain; PN: peripheral neuropathy.

**Figure 2 jcm-12-01483-f002:**
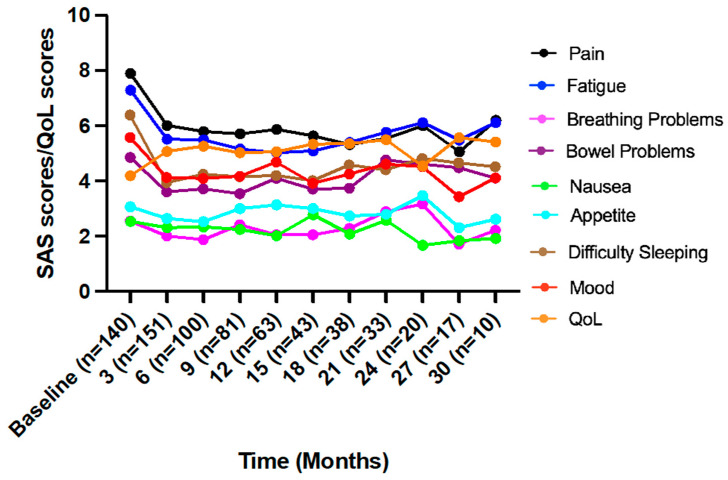
Mean symptom assessment scale scores for all patients assessing symptom distress (10 = worst) and quality of life scores (10 = best). Analysis was possible to 30 months. *n*: number of patients contributing data at each timepoint; QoL: Quality of life; SAS: Symptom Assessment Scale.

**Figure 3 jcm-12-01483-f003:**
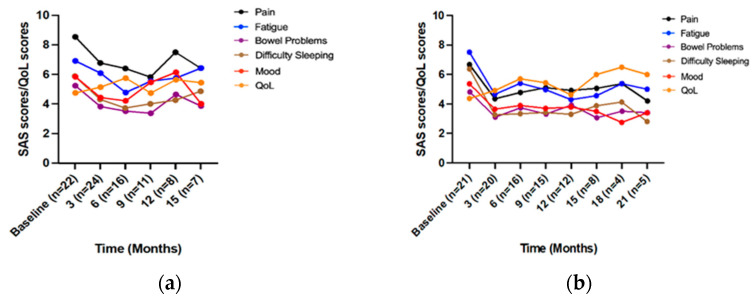
Mean symptom assessment scale scores for patients with a condition assessing symptom distress (10 = worst) and quality of life scores (10 = best). (**a**) Spondylosis (to 15 months); (**b**) Parkinson’s disease (to 21 months); (**c**) Chronic pain syndrome (to 21 months); (**d**) Combined neuropathic pain and peripheral neuropathy (to 21 months); (**e**) Migraine (to 18 months); (**f**) Multiple sclerosis (to 15 months). *n* = number of patients contributing data at each timepoint; QoL: Quality of life; SAS: Symptom Assessment Scale.

**Figure 4 jcm-12-01483-f004:**
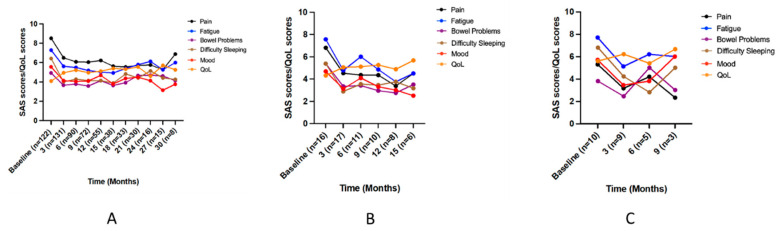
Mean symptom assessment scale scores for patients with an indication assessing symptom distress (10 = worst) and quality of life scores (10 = best). (**A**) Pain (to 30 months); (**B**) Muscle spasm (to 15 months); (**C**) Sleep (to 9 months). *n*: number of patients contributing data at each timepoint; QoL: Quality of life; SAS: Symptom Assessment Scale.

**Figure 5 jcm-12-01483-f005:**
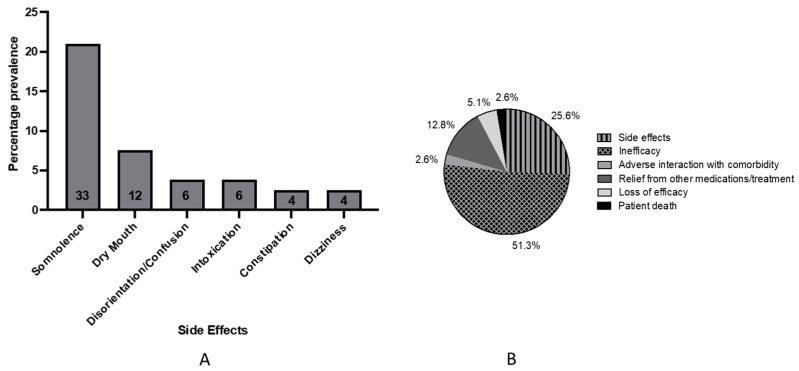
Safety and treatment status: (**A**) Prevalence of the six most common side effects (*n* = 65). The number of patients who reported each side effect is displayed within each column and percentages calculated from total population (*n* = 157); (**B**) Reasons for treatment cessation (*n* = 39). Inefficacy = no effect on condition/indication, Loss of efficacy = initial effect wore off over time.

**Table 1 jcm-12-01483-t001:** Clinical characteristics of adults prescribed medicinal cannabis at the clinic.

Characteristics	Number of Patients (%) (*n* = 157)
Age (years)	
Mean ± standard deviation	63.0 ± 16.0
Range	18–100
Over 40 years	141 (89.8)
Under 65 years	75 (47.8)
65 years and over	82 (52.2)
Gender ^a^	
Male	57 (36.3)
Female	100 (63.7)
Place of residence	
Greater Sydney Region ^b^	136 (86.6)
Non-Sydney NSW	17 (10.8)
Queensland	1 (0.6)
Australian Capital Territory	1 (0.6)
Tasmania	1 (0.6)
Victoria	1 (0.6)
Country of birth	
Australia	105 (66.9)
Country other than Australia	42 (26.7)
Unknown	10 (6.4)
Indigenous status	
Aboriginal or Torres Strait Islander	0 (0)
Not Indigenous	153 (97.5)
Unknown	4 (2.5)
Relationship status	
Single ^c^	49 (31.2)
Partnered ^d^	90 (57.3)
Unknown	18 (11.5)
Previous cannabis use	
Yes	28 (17.8)
No	1 (0.6)
Not stated	128 (81.5)
**Condition class ^e^ and** condition	
**Musculoskeletal**	39 (24.8) **^f^**
Spondylosis	25 (15.9)
**Neurological**	91 (58.0)
Parkinson’s disease	23 (14.6)
Migraine	16 (10.2)
Multiple sclerosis	10 (6.4)
Neuropathic pain	9 (5.7)
Peripheral neuropathy	9 (5.7)
**Autoimmune**	14 (8.9)
**Inflammatory**	10 (6.4)
**Other**	23 (14.6)
Chronic pain syndrome	19 (12.1)
Number of conditions treated	
One	137 (87.3)
Two	19 (12.1)
Three	1 (0.6)
Indications	
Pain	136 (86.6)
Muscle spasms	18 (11.5)
Sleep	10 (6.4)
**Cannabis-based medicines used**	
**Number**	
1	112 (71.3) **^g^**
2	31 (19.7)
≥3	14 (8.9)
Type (route of administration)	
Oil (oral)	156 (99.4)
Flower (inhalation)	5 (3.2)
Oromucosal spray (sublingual)	1 (0.6)
Cannabinoid ratios	
CBD only ^h^	22 (14.0)
CBD dominant ^i^	5 (3.2)
Low Δ-9-THC ^j^	12 (7.6)
Balanced ^k^	133 (84.7)
Δ-9-THC dominant/Δ-9-THC only ^l^	15 (9.6)
**Cannabinoid highest dosage (mg/day)**	**Mean ± SD**
CBD	34.8 ± 48.04 *
Δ-9-THC	16.9 ± 17.23

CBD: Cannabidiol. Δ-9-THC: Δ-9-tetrahydrocannabinol. ^a^ As reported in the medical notes; ^b^ Covers around 12,368.2 km^2^ and 37.6% (*n* = 59) of patients travelled 20 km or more to access the clinic; ^c^ Not in a relationship, including divorced, widowed, separated; ^d^ In a relationship, including married, de facto, same sex; **^e^** For each condition class, the conditions with ≥10 patients are shown (neuropathic pain and peripheral neuropathy have been combined in the analysis); **^f^** Percentages for ‘Condition class’ and ‘Indications’ are given as individual frequencies out of the entire study population, as some patients had more than one condition/indication. Percentages will add to over 100; **^g^** Percentages for ‘CBM type’ and ‘Cannabinoid ratios’ are given as individual frequencies out of the entire study population, as some patients used more than one type and/or more than one ratio percentages will add to more than 100; ^h^ Ratios (Δ-9-THC:CBD) included 0:25, 0:100. ^i^ Ratios included 1:20, 2:25; ^j^ Ratios included 5:20; ^k^ Ratios included 10:10, 25:25, 12.5:12.5, 2.7:2.5, 2.5:3.75; ^l^ Ratios included 10:0, 20:1, 18:1, 2.5:1, 25:2. * Significantly higher *p* < 0.0001.

**Table 2 jcm-12-01483-t002:** Symptom assessment scale scores for all patients.

	Pain			Fatigue			Breathing Problems			Bowel Problems			Nausea			Appetite			Difficulty Sleeping			Mood			QoL		
*p* Value, F (DFn, DFd)	<0.0001, F (3.7, 342) = 36			<0.0001, F (3.8, 357) = 28			0.0035, F (3.8, 351) = 4.1			<0.0001, F (3.7, 340) = 8.3			0.1397, F (3.7, 340) = 1.8			0.0465, F (3.5, 327) = 2.5			<0.0001, F (3.1, 292) = 40			<0.0001, F (3.3, 303) = 14			<0.0001, F (3.5, 313) = 7.5		
	M	*p*-Value	95% CI of Diff.	M	*p*-Value	95% CI of Diff.	M	*p*-Value	95% CI of Diff.	M	*p*-Value	95% CI of Diff.	M	*p*-Value	95% CI of Diff.	M	*p*-Value	95% CI of Diff.	M	*p*-Value	95% CI of Diff	M	*p*-Value	95% CI of Diff	M	*p*-Value	95% CI of Diff
**Baseline** **(*n* = 140)**	7.9			7.3			2.5			4.9			2.5			3.1			6.4			5.6			4.2		
**3 months** **(*n* = 151)**	** *6.0* **	<0.0001	1.4 to 2.3	** *5.5* **	<0.0001	1.2 to 2.3	** *2.0* **	0.0165	0.076 to 1.0	** *3.6* **	<0.0001	0.58 to 1.9	2.3	0.7853	−0.38 to 0.84	2.6	0.1852	−0.13 to 0.98	** *3.9* **	<0.0001	1.9 to 3.0	** *4.1* **	<0.0001	0.93 to 2.0	** *5.1* **	<0.0001	−1.3 to −0.45
**6 months** **(*n* = 100)**	** *5.8* **	<0.0001	1.5 to 2.7	** *5.5* **	<0.0001	1.3 to 2.4	** *1.9* **	0.0084	0.14 to 1.2	** *3.7* **	0.004	0.30 to 2.0	2.3	0.8621	−0.43 to 0.83	2.5	0.2511	−0.23 to 1.3	** *4.2* **	<0.0001	1.6 to 2.7	** *4.1* **	<0.0001	0.86 to 2.1	** *5.3* **	0.0001	−1.7 to −0.47
**9 months** **(*n* = 81)**	** *5.7* **	<0.0001	1.6 to 2.8	** *5.1* **	<0.0001	1.4 to 2.8	2.4	0.9486	−0.45 to 0.72	** *3.5* **	0.0015	0.43 to 2.2	2.2	0.7263	−0.43 to 1.0	3.0	0.9983	−0.71 to 0.86	** *4.1* **	<0.0001	1.5 to 2.9	** *4.2* **	<0.0001	0.70 to 2.1	** *5.0* **	0.0039	−1.4 to −0.22
**12 months** **(*n* = 63)**	** *5.9* **	<0.0001	1.4 to 2.7	** *5.0* **	<0.0001	1.5 to 3.0	2.0	0.1381	−0.11 to 1.1	4.1	0.0947	−0.09 to 1.6	2.0	0.3074	−0.27 to 1.3	3.1	0.9997	−1.1 to 0.95	** *4.2* **	<0.0001	1.2 to 3.1	** *4.7* **	0.0244	0.090 to 1.7	** *5.1* **	0.0221	−1.7 to −0.1

Analysis was possible to 12 months. Bolded, italicized, and highlighted means are statistically significant compared to baseline at *p* < 0.05 level as analyzed by Dunnett’s test. Abbreviations: CI, Confidence interval; DFn, degree of freedom for the numerator of the F ratio; DFd, degree of freedom for the denominator of the F ratio; Diff., difference; M, Mean; QoL, Quality of life.

**Table 3 jcm-12-01483-t003:** Symptom assessment scale scores for patients with the condition: (a) spondylosis; (b) Parkinson’s disease; (c) chronic pain syndrome; (d) neuropathic pain and peripheral neuropathy; (e) migraine; (f) multiple sclerosis.

**Spondylosis ^a^**	**Pain**			**Fatigue**			**Bowel Problems**			**Difficulty Sleeping**			**Mood**			**Quality of Life**		
***p*-Value,** **F (DFn, DFd)**	0.0001, F (2.3, 35) = 11			0.0039, F (2.0, 31) = 6.7			0.0260, F (1.8, 28) = 4.3			0.0037, F (1.7, 26) = 7.5			0.0192, F (2.0, 30) = 4.5			0.3216, F (2.9, 43) = 1.2		
	**Mean**	***p*-Value**	**95% CI of Diff.**	**Mean**	***p*-Value**	**95% CI of Diff.**	**Mean**	***p*-Value**	**95% CI of Diff.**	**Mean**	***p*-Value**	**95% CI of Diff.**	**Mean**	***p*-Value**	**95% CI of Diff.**	**Mean**	***p*-Value**	**95% CI of Diff.**
**Baseline** **(*n* = 22)**	8.5			6.9			5.2			5.9			5.8			4.7		
**3 months** **(*n* = 24)**	** *6.8* **	0.0001	0.92 to 2.6	6.1	0.1431	−0.22 to 1.9	3.8	0.1635	−0.44 to 3.2	** *4.3* **	0.0079	0.39 to 2.7	** *4.4* **	0.0292	0.13 to 2.7	5.1	0.7412	−1.5 to 0.75
**6 months** **(*n* = 16)**	** *6.4* **	0.0005	1.1 to 3.2	** *4.8* **	0.0039	0.72 to 3.5	3.5	0.2181	−0.80 to 4.2	** *3.7* **	0.0016	0.88 to 3.4	** *4.2* **	0.0295	0.16 to 3.1	5.7	0.3092	−2.7 to 0.68
**9 months** **(*n* = 11)**	** *5.8* **	0.0044	1.1 to 4.4	5.5	0.1269	−0.38 to 3.1	3.4	0.2046	−0.91 to 4.6	** *4* **	0.0075	0.59 to 3.1	5.5	0.9243	−1.7 to 2.5	4.7	>0.9999	−1.8 to 1.8
**Parkinson’s disease ^b^**	**Pain**			**Fatigue**			**Bowel problems**			**Difficulty sleeping**			**Mood**			**Quality of life**		
***p*-Value,** **F (DFn, DFd)**	0.0068, F (3.4, 49) = 4.3			0.0005, F (2.9, 40) = 7.4			0.2521, F (2.1, 30) = 1.4			0.0002, F (1.6, 22) = 15			0.1228, F (2.3, 32) = 2.2			0.2160, F (2.5, 34) = 1.6		
	**Mean**	***p*-Value**	**95% CI of diff.**	**Mean**	***p*-Value**	**95% CI of diff.**	**Mean**	***p*-Value**	**95% CI of diff.**	**Mean**	***p*-Value**	**95% CI of diff.**	**Mean**	***p*-Value**	**95% CI of diff.**	**Mean**	***p*-Value**	**95% CI of diff.**
**Baseline** **(*n* = 21)**	6.7			7.5			4.8			6.4			5.4			4.4		
**3 months** **(*n* = 20)**	** *4.4* **	0.0039	0.71 to 3.9	** *4.7* **	0.0002	1.4 to 4.4	3.1	0.0615	−0.068 to 3.5	** *3.3* **	<0.0001	1.8 to 4.4	** *3.7* **	0.0214	0.23 to 3.2	4.9	0.5842	−1.7 to 0.65
**6 months** **(*n* = 16)**	4.8	0.0793	−0.19 to 4.0	** *5.4* **	0.0161	0.38 to 3.9	3.7	0.6584	−1.6 to 3.8	** *3.3* **	0.0001	1.7 to 4.4	3.9	0.0707	−0.11 to 3.1	5.7	0.1724	−3.1 to 0.46
**9 months** **(*n* = 15)**	5.1	0.0917	−0.22 to 3.4	** *5* **	0.0067	0.73 to 4.4	3.3	0.1953	−0.59 to 3.6	** *3.4* **	0.0007	1.4 to 4.5	3.7	0.0791	−0.17 to 3.5	5.4	0.3543	−2.9 to 0.78
**12 months** **(*n* = 12)**	4.9	0.1614	−0.58 to 4.1	** *4.3* **	0.0092	0.85 to 5.6	3.9	0.5466	−1.1 to 2.9	** *3.3* **	0.0235	0.42 to 5.8	3.8	0.1499	−0.48 to 3.7	4.6	0.9954	−2.7 to 2.2
**Chronic pain syndrome ^c^**	**Pain**			**Fatigue**			**Bowel problems**			**Difficulty sleeping**			**Mood**			**Quality of life**		
***p*-Value,** **F (DFn, DFd)**	0.0012, F (2.1, 23) = 8.8			0.0362, F (1.7, 19) = 4.2			0.7399, F (2.4, 26) = 0.36			0.0128, F (2.6, 29) = 4.6			0.0013, F (2.5, 27) = 7.5			0.1453, F (2.2, 22) = 2.1		
	**Mean**	***p*-Value**	**95% CI of diff.**	**Mean**	***p*-Value**	**95% CI of diff.**	**Mean**	***p*-Value**	**95% CI of diff.**	**Mean**	***p*-Value**	**95% CI of diff.**	**Mean**	***p*-Value**	**95% CI of diff.**	**Mean**	***p*-Value**	**95% CI of diff.**
**Baseline** **(*n* = 16)**	9.1			7.1			5.1			6.4			6.1			3.7		
**3 months** **(*n* = 19)**	** *6.8* **	0.0052	0.67 to 3.9	5.3	0.2321	−0.80 to 4.3	4.3	0.8853	−2.4 to 4.1	** *3.4* **	0.0015	1.2 to 4.9	** *3.9* **	0.01	0.50 to 3.9	** *5.1* **	0.0293	−2.6 to −0.13
**6 months** **(*n* = 11)**	** *7.1* **	0.043	0.063 to 3.8	6.5	0.6122	−0.94 to 2.2	4.5	0.94	−2.6 to 3.9	5.4	0.582	−1.5 to 3.7	4.2	0.1026	−0.39 to 4.2	4	0.9957	−2.7 to 2.2
**9 months** **(*n* = 11)**	** *5.9* **	0.0031	1.3 to 5.0	** *4.2* **	0.0147	0.67 to 5.1	4.5	0.9323	−2.4 to 3.6	3.7	0.1257	−0.74 to 6.2	4	0.1444	−0.70 to 4.9	5.2	0.2037	−3.8 to 0.83
**12 months** **(*n* = 10)**	** *7.8* **	0.0348	0.11 to 2.5	6.1	0.6067	−1.7 to 3.7	5.3	0.9998	−4.3 to 4.0	4.6	0.5133	−2.5 to 6.2	7.6	0.4838	−4.7 to 1.8	4	0.9806	−2.1 to 1.6
**Neuropathic** **pain/peripheral neuropathy ^d^**	**Pain**			**Fatigue**			**Bowel problems**			**Difficulty sleeping**			**Mood**			**Quality of life**		
***p*-Value,** **F (DFn, DFd)**	<0.0001, F (2.5, 27) = 14			0.0326, F (2.5, 26) = 3.6			0.0563, F (2.6, 29) = 2.9			0.0046, F (2.2, 24) = 6.4			0.5335, F (2.0, 22) = 0.65			0.0771, F (2.7, 29) = 2.6		
	**Mean**	***p*-Value**	**95% CI of diff.**	**Mean**	***p*-Value**	**95% CI of diff.**	**Mean**	***p*-Value**	**95% CI of diff.**	**Mean**	***p*-Value**	**95% CI of diff.**	**Mean**	***p*-Value**	**95% CI of diff.**	**Mean**	***p*-Value**	**95% CI of diff.**
**Baseline** **(*n* = 17)**	8.8			6.6			4.4			6.8			4.6			4.2		
**3 months** **(*n* = 18)**	** *6.2* **	<0.0001	1.5 to 3.8	4.5	0.1047	−0.35 to 4.5	** *3* **	0.0273	0.14 to 2.7	** *4.1* **	0.0014	1.1 to 4.4	3.9	0.7809	−1.4 to 2.8	** *5.4* **	0.0353	−2.4 to −0.073
**6 months** **(*n* = 12)**	** *5.2* **	<0.0001	2.2 to 5.1	5	0.153	−0.53 to 3.8	3	0.422	−1.3 to 4.1	** *4.3* **	0.0073	0.74 to 4.4	4.2	0.9828	−2.1 to 2.8	5.7	0.096	−3.4 to 0.25
**9 months** **(*n* = 9)**	** *6.1* **	0.0038	1.1 to 4.4	4.4	0.209	−1.3 to 5.7	2.3	0.3018	−1.5 to 5.7	4.5	0.0616	−0.12 to 4.8	4.2	0.943	−1.9 to 2.7	5.2	0.3717	−3.0 to 0.94
**12 months** **(*n* = 10)**	** *6.2* **	0.008	0.79 to 4.5	4.1	0.0949	−0.48 to 5.6	2.7	0.2241	−0.90 to 4.4	5	0.1147	−0.43 to 4.2	3.1	0.1474	−0.49 to 3.6	5.3	0.4064	−3.2 to 1.0
**Migraine ^e^**	**Pain**			**Fatigue**			**Bowel problems**			**Difficulty sleeping**			**Mood**			**Quality of life**		
***p*-Value,** **F (DFn, DFd)**	0.2182, F (1.7, 19) = 1.7			0.0144, F (2.0, 23) = 5.1			0.1243, F (2.3, 27) = 2.2			0.0171, F (2.4, 27) = 4.4			0.2191, F (1.7, 18) = 1.7			0.1138, F (2.0, 20) = 2.4		
	**Mean**	***p*-Value**	**95% CI of diff.**	**Mean**	***p*-Value**	**95% CI of diff.**	**Mean**	***p*-Value**	**95% CI of diff.**	**Mean**	***p*-Value**	**95% CI of diff.**	**Mean**	***p*-Value**	**95% CI of diff.**	**Mean**	***p*-Value**	**95% CI of diff.**
**Baseline** **(*n* = 17)**	7.9			6.9			4.3			6.4			5.7			3.8		
**3 months** **(*n* = 18)**	7.2	0.7377	−1.4 to 2.7	** *5.8* **	0.0287	0.12 to 2.2	3.4	0.4631	−0.97 to 2.8	4.2	0.0536	−0.030 to 4.4	4.8	0.2354	−0.46 to 2.3	4.6	0.3831	−2.4 to 0.72
**6 months** **(*n* = 12)**	6.9	0.1592	−0.33 to 2.3	** *5.5* **	0.0424	0.049 to 2.8	3	0.418	−1.3 to 4.0	3.9	0.0606	−0.11 to 5.1	5.3	0.909	−1.9 to 2.8	4.8	0.2834	−2.7 to 0.68
**9 months** **(*n* = 9)**	6.4	0.2864	−1.0 to 3.9	5.8	0.3116	−0.85 to 3.1	4.1	0.9874	−2.8 to 3.3	4.3	0.1356	−0.64 to 4.9	4.3	0.546	−2.0 to 4.8	4.2	0.9356	−2.8 to 2.0
**Multiple sclerosis ^f^**	**Pain**			**Fatigue**			**Bowel problems**			**Difficulty sleeping**			**Mood**			**Quality of life**		
	**Median**	***p*-Value**	**95% CI of diff.**	**Median**	***p*-Value**	**95% CI of diff.**	**Median**	***p*-Value**	**95% CI of diff.**	**Median**	***p*-Value**	**95% CI of diff.**	**Median**	***p*-Value**	**95% CI of diff.**	**Median**	***p*-Value**	**95% CI of diff.**
**Baseline** **(*n* = 10)**	9			9.5			3			7.5			7			3.5		
**3 months** **(*n* = 10)**	6.3	0.0859	−3.0 to 0.0	** *6* **	0.0039	−6.0 to −1.0	2.5	0.4609	−5.0 to 1.0	** *4.3* **	0.0039	−5.5 to −1.0	** *5* **	0.0195	−5.0 to 0.50	5	0.0508	0.0 to 3.0

Analysis was possible to 9 ^a,e^ and 12 ^b–d^ months. Bolded, italicized, and highlighted means are statistically significant compared to baseline at *p* < 0.05 level as analyzed by Dunnett’s test. ^f^ Analysis was possible to 3 months. Bolded, italicized, and highlighted medians are statistically significant compared to baseline at *p* < 0.05 level as analyzed by Wilcoxon signed-rank test. Abbreviations: CI, Confidence interval; DFn, degree of freedom for the numerator of the F ratio; DFd, degree of freedom for the denominator of the F ratio; Diff., difference.

**Table 4 jcm-12-01483-t004:** Symptom assessment scale scores for patients with the indication: (a) pain; (b) muscle spasm; (c) sleep.

**Pain ^a^**	**Pain**			**Fatigue**			**Bowel Problems**			**Difficulty Sleeping**			**Mood**			**Quality of Life**		
***p*-Value,** **F (DFn, DFd)**	**<0.0001,** **F (5.0, 308) = 26**			**<0.0001,** **F (5.6, 342) = 15**			**<0.0001,** **F (5.2, 316) = 5.3**			**<0.0001,** **F (4.8, 296) = 21**			**<0.0001,** **F (5.0, 305) = 6.9**			**0.0004,** **F (4.7, 279) = 4.8**		
	**Mean**	***p*-Value**	**95% CI of Diff.**	**Mean**	***p*-Value**	**95% CI of Diff.**	**Mean**	***p*-Value**	**95% CI of Diff.**	**Mean**	***p*-Value**	**95% CI of Diff.**	**Mean**	***p*-Value**	**95% CI of Diff.**	**Mean**	***p*-Value**	**95% CI of Diff.**
**Baseline** **(*n* = 122)**	8.5			7.3			4.9			6.4			5.6			4.1		
**3 months** **(*n* = 131)**	** *6.5* **	<0.0001	1.5 to 2.5	** *5.6* **	<0.0001	1.0 to 2.3	** *3.7* **	0.0001	0.50 to 2.0	** *4* **	<0.0001	1.8 to 3.0	** *4.1* **	<0.0001	0.83 to 2.0	** *4.9* **	<0.0001	−1.3 to −0.40
**6 months** **(*n* = 90)**	** *6.1* **	<0.0001	1.9 to 3.0	** *5.5* **	<0.0001	1.2 to 2.4	** *3.8* **	0.0121	0.18 to 2.1	** *4.3* **	<0.0001	1.5 to 2.8	** *4* **	<0.0001	0.78 to 2.2	** *5.2* **	0.0003	−1.8 to −0.42
**9 months** **(*n* = 72)**	** *6* **	<0.0001	1.8 to 3.1	** *5.2* **	<0.0001	1.3 to 2.9	** *3.6* **	0.0049	0.31 to 2.4	** *4.1* **	<0.0001	1.5 to 3.1	** *4.1* **	<0.0001	0.63 to 2.3	** *5* **	0.0137	−1.6 to −0.13
**12 months** **(*n* = 55)**	** *6.2* **	<0.0001	1.6 to 3.0	** *5* **	<0.0001	1.4 to 3.1	4.1	0.1599	−0.18 to 1.8	** *4.1* **	<0.0001	1.2 to 3.4	4.7	0.092	−0.088 to 1.8	** *5.1* **	0.029	−2.0 to −0.074
**Muscle spasm ^a^**	**Pain**			**Fatigue**			**Bowel problems**			**Difficulty sleeping**			**Mood**			**Quality of life**		
***p*-Value,** **F (DFn, DFd)**	0.0490, F (2.8, 28) = 3.1			0.0043, F (1.9, 19) = 7.6			0.1481, F (2.6, 26) = 2.0			0.0555, F (1.4, 14) = 3.9			0.1790, F (2.0, 20) = 1.9			0.4053, F (2.2, 21) = 0.96		
	**Mean**	***p*-Value**	**95% CI of diff.**	**Mean**	***p*-Value**	**95% CI of diff.**	**Mean**	***p*-Value**	**95% CI of diff.**	**Mean**	***p*-Value**	**95% CI of diff.**	**Mean**	***p*-Value**	**95% CI of diff.**	**Mean**	***p*-Value**	**95% CI of diff.**
**Baseline** **(*n* = 16)**	6.8			7.6			5.4			5.4			4.7			4.3		
**3 months** **(*n* = 17)**	** *4.5* **	0.0201	0.33 to 4.2	** *4.8* **	0.0003	1.4 to 4.2	3.3	0.0978	−0.31 to 4.4	** *2.9* **	0.0028	0.89 to 4.1	3.1	0.1239	−0.34 to 3.5	5	0.3463	−2.0 to 0.51
**6 months** **(*n* = 11)**	4.4	0.1706	−0.90 to 5.8	6	0.1522	−0.50 to 3.6	3.4	0.1528	−0.63 to 4.6	** *3.5* **	0.0174	0.34 to 3.3	4.1	0.8001	−1.4 to 2.6	5.1	0.7777	−3.3 to 1.8
**9 months** **(*n* = 10)**	4.4	0.1326	−0.69 to 5.6	** *4.9* **	0.0266	0.34 to 5.1	3	0.1514	−0.80 to 5.7	3.5	0.094	−0.32 to 4.2	3.3	0.3255	−1.0 to 3.8	5.3	0.5827	−3.2 to 1.3
**12 months (*n* = 8)**	3.4	0.1348	−1.2 to 8.1	** *3.8* **	0.0072	1.5 to 6.2	2.8	0.1428	−0.99 to 6.2	3.8	0.6223	−3.1 to 6.3	3	0.2172	−1.0 to 4.4	4.9	0.9376	−3.9 to 2.8
**Sleep ^b^**	**Pain**			**Fatigue**			**Bowel problems**			**Difficulty sleeping**			**Mood**			**Quality of life**		
	**Median**	***p*-Value**	**95% CI of diff.**	**Median**	***p*-Value**	**95% CI of diff.**	**Median**	***p*-Value**	**95% CI of diff.**	**Median**	***p*-Value**	**95% CI of diff.**	**Median**	***p*-Value**	**95% CI of diff.**	**Median**	***p*-Value**	**95% CI of diff.**
**Baseline** **(*n* = 9)**	7			8			3			7.5			7			5		
**3 months** **(*n* = 9)**	2	0.1406	−5.0 to 0.50	** *5.5* **	0.0391	−6.0 to 0.0	2	0.5625	−3.0 to 2.5	** *4* **	0.0117	−4.0 to −1.5	** *4* **	0.0156	−4.5 to 0.50	6	0.3281	−1.0 to 3.0

^a^ Analysis was possible to 12 months. Bolded, italicized, and highlighted means are statistically significant compared to baseline at *p* < 0.05 level as analyzed by Dunnett’s test. ^b^ Analysis was possible to 3 months. Bolded, italicized, and highlighted medians are statistically significant compared to baseline at *p* < 0.05 level as analyzed by Wilcoxon signed-rank test. Abbreviations: CI, Confidence interval; DFn, degree of freedom for the numerator of the F ratio; DFd, degree of freedom for the denominator of the F ratio; Diff., difference.

## Data Availability

De-identified data that support the findings of this study are available on request from the corresponding author. The data are not publicly available due to privacy or ethical restrictions.

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
