# Peer review of "A Retrospective Medical Record Review of Adults with Non-Cancer Diagnoses Prescribed Medicinal Cannabis"

_jcm, 2023, doi:10.3390/jcm12041483_

Round 1

Reviewer 1 Report

This paper discusses the experience of a single medical cannabis prescriber from February 01 2018 to November 30 2021 at the private Jewish Wolper Hospital in Eastern Sydney.

 General Background Situation

Its general perspective must be set against the overall picture of what has occurred in Australia with medical cannabis prescribing in recent years.  Cannabis has been made available for doctors to prescribe to selected patients in Australia as mentioned in the article during this period.  The commonest pattern is that patient with a history of cannabis use utilize legal access to cannabis as a route to accessing their drug of choice which cannot be interfered with by law enforcement agencies.  Various symptoms such as pain, anxiety, depression or nausea become the “ticket” by which such patients gain access to cannabis.  Of course these symptoms commonly arise in the process of cannabis dependence or withdrawal.   That is to say that the symptoms are nominal only – the real issue is the management of cannabis dependence in patients who are able to afford its not inconsiderable fee structure.  

This somewhat ambiguous and confusing legal situation has now evolved to the point where law enforcement itself advises patients caught with cannabis to “go and get a [legal] prescription”.

 Moreover the whole medical cannabis industry in Australia has arisen in reverse.  In this nation cannabis is something of “an [alleged] cure looking for a disease”.  This is the reverse of the usual process where safety and efficacy – and of course principal indication for use - are all worked out meticulously before the drug comes to market and post-marketing surveillance is then instituted.

At this time the industry generally is therefore looking hard to justify and legitimate its own existence.

Cannabis Withdrawal

This background is important for two reasons.   Firstly many of the patients accessing legal cannabis are actually cannabis dependent and therefore cannabis tolerant.  Therefore calculations of side effects in this cannabis tolerant population are very different to what might be expected in a cannabis naïve population.  Secondly when cannabis is denied to cannabis dependent patients they experience many symptoms such as pain (in muscles and joints and head), psychological symptoms (anxiety, agitation, insomnia, depression, hallucinations and psychosis), physical symptoms (tremor, weakness, tiredness) and gastrointestinal symptoms (nausea, loss of appetite, abdominal cramps and pains, diarrhoea).

That is it is the cannabis dependency which is inducing the symptomatology.  To argue then cannabis reduces the symptoms of cannabis withdrawal is facile and of minimal medical interest. 

If one reviews this list it will be observed that it reads like the list of indications for which cannabis was given by these authors. 

There is also the issue of the link between cannabis and mental illness which is a major issue in any study considering safety concerns.  A simple Google or PubMed search will demonstrate that cannabis has been definitively shown to be linked with many major mental disorders including anxiety, depression, bipolar disorder, schizophrenia, and suicidal thoughts and actions.  Numerous (more than 70) recent mass shootings in USA have also been conducted by individuals under the influence of cannabis which effectively links homicide as well as suicide with cannabis intoxication and / or withdrawal. 

Many studies have also shown elevated mortality amongst cannabis users.  This is clearly a traditional “hard end point” which cannot – and should not - be lightly or frivolously overlooked.

In addition there is the now many studies linking cannabis with major genotoxic outcomes including cancer, birth defects and aging as has recently been described.

The present study therefore has many profound flaws.

Cannabis Non-Naivete

From the above remarks a central concern is the number of clients which had used cannabis prior to study entry. 

Importantly the authors make the patently false claim that “only 17.8% stating previous cannabis use (Table 1).” [Manuscript Line 95]

This important remark is obviously false and this gross untruth by itself becomes a fatal flaw in the paper.

If one does as directed and refers to Table 1 one finds indeed that 28 patients had used cannabis previously, one patient had not AND THE DATA FOR THE OTHER 128 PATIENTS WAS MISSING!

This has two corollaries.  28/29 = 96.55% of patients had used cannabis prior where data exists.  That is to say that where data exists the overwhelming majority – and indeed virtually all - of these patients were known to be cannabis dependent.  This conclusion is the exact reverse of the one presented by the study authors.

Since the issue of prior cannabis experience / pre-existing cannabis tolerance is likely the major question of the study upon which turns all the issues of interpreting both cannabis efficacy and safety it thus bears close examination.

The authors state that 217 patients were studied.

The authors state that the data from 50 was deficient. This is worrying in itself as it suggests that the note taking in these records was deficient in important respects in 50/217 or 23.04% of clinical files.

The authors further mention in Table 1 that 128 patients had missing data on the prior use issue making 178/217 patients with missing data relating to prior cannabis use which is a missing data rate of 86.6%.

This extremely high error rate is completely unacceptable and makes the whole study uninterpretable and / or essentially of no worth.

Moreover there are several hints or suggestions in the paper that cannabis use in this population was significant including:

i)                   The study was conducted in the eastern suburbs of Sydney which are socioeconomically generally (extremely) privileged in an area where drug use in general and cannabis use in particular is common;

ii)                 The rate of side effects from cannabis was very much lower (about 21%) than is widely reported elsewhere after the use of marinol and nabilone where tiredness, nausea and hallucinations are widely reported to occur in from 70-100% of the cannabis naïve users, a major issue which therefore implies a high degree of pre-existing tolerance amongst the present patient cohort;

iii)               As mentioned above the known rate of prior cannabis use where data is available was actually 96.55%.

The study population was not representative of the general Australian population in several ways:

i)                   The study was conducted in a Private hospital which immediately sets it outside of mainstream Australian society

ii)                 The study was conducted in the socioeconomically exclusive eastern suburbs of Sydney

iii)               Patients with known mental illness were referred elsewhere (Line 289)

iv)               Patients with known sleep disorders were referred elsewhere (Line 289).

 This is really extraordinary!  The study group was tightly selected on both socioeconomic and sociodemographic criteria by both geography, economic standing and possibly ethnicity.  And yet even from this highly select group “hard cases” were excluded!

The exclusion of cases of mental illness invalidates all of the authors’ comments on mental illness including depression and anxiety.

Indeed Line 289 also states that patient with sleep disorders were also rejected by the prescribing physician.  This remark likely invalidates and indeed directly contradicts all the remarks on sleep and insomnia in this study.

The claims relating to so-called cannabis efficacy in relieving various symptoms clearly leaves open the possibility that the symptoms which were being relieved had actually arisen due to long standing cannabis dependence.  The inability to accurately ascertain or interpret prior cannabis use history invalidates all of these claims and reduces them to the very trivial likelihood that the physician was only treating the symptoms of cannabis dependence.

The claims made relating to alleged cannabis safety are all grossly erroneous due to the possibility and indeed probability that the study population was largely cannabis dependent.

Moreover global and generic claims relating to alleged cannabis safety are wildly inflated in such a highly select population – and in any case are grossly irresponsible when issues as major as cancer, teratogenicity and ageing following cannabis exposure have been seriously raised.

For these reasons this study cannot be recommended in any respect.  Its severely limiting 86.6% missing data rate on the key question considered completely invalidates and vitiates all the comments relating to alleged efficacy such that it is likely reporting effects in a largely cannabis dependent population and leads only to the trivial and non-reportable suggestion that cannabis administration is an effective treatment for cannabis dependence.  Similarly the alleged low rate of side effects is also likely an artefact of that same cannabis tolerance in the study group.

To claim that cannabis use is safe, in the face of a literature replete with mental, cardiac respiratory, immunological, reproductive, cancer, aging, genotoxic and epigenotoxic complications of cannabis in both the exposed and in subsequent generations is either misleading and overtly fraudulent, or in the alternative, overtly and unacceptably uninformed.  

Like the fledgling cannabis industry itself this paper reads like a cannabis advertorial seeking to justify its own existence.  Its uninformed, facile and grossly incomplete treatment both of its study material and the wider subject itself should preclude it from further serious editorial consideration.

Reviewer 2 Report

In this retrospective study, the authors examined the use of cannabis to treat non-cancer conditions.  In short, their results showed that prescribed cannabis can be used for a variety of conditions with safety.  The manuscript is well-written.  There are several points that can easily be addressed:

1. Table 1 can be simplified and made shorter.  Every condition is listed even if there was only one patient.  For example, autoimmune and inflammatory can be stand alone without listing each disease.

2. Tables 2, 3, and 4 are complicated to read.  The data are presented in grafts that are much easier to understand.  The authors should delete these tables or include them in supplementary data.
